# Forming Big Datasets through Latent Class Concatenation of Imperfectly Matched Databases Features

**DOI:** 10.3390/genes10090727

**Published:** 2019-09-19

**Authors:** Christopher W. Bartlett, Brett G. Klamer, Steven Buyske, Stephen A. Petrill, William C. Ray

**Affiliations:** 1Battelle Center for Mathematical Medicine, Abigail Wexner Research Institute, Nationwide Children’s Hospital, Columbus, OH 43215, USA; 2Department of Pediatrics, College of Medicine, The Ohio State University, Columbus, OH 43215, USA; 3Departments of Statistics and Genetics, Rutgers University, Piscataway, NJ 08854, USA; 4Department of Psychology, College of Arts and Sciences, The Ohio State University, Columbus, OH 43210, USA

**Keywords:** data integration, databases, informatics, data blending, data pools

## Abstract

Informatics researchers often need to combine data from many different sources to increase statistical power and study subtle or complicated effects. Perfect overlap of measurements across academic studies is rare since virtually every dataset is collected for a unique purpose and without coordination across parties not-at-hand (i.e., informatics researchers in the future). Thus, incomplete concordance of measurements across datasets poses a major challenge for researchers seeking to combine public databases. In any given field, some measurements are fairly standard, but every organization collecting data makes unique decisions on instruments, protocols, and methods of processing the data. This typically denies literal concatenation of the raw data since constituent cohorts do not have the same measurements (i.e., columns of data). When measurements across datasets are similar prima facie, there is a desire to combine the data to increase power, but mixing non-identical measurements could greatly reduce the sensitivity of the downstream analysis. Here, we discuss a statistical method that is applicable when certain patterns of missing data are found; namely, it is possible to combine datasets that measure the same underlying constructs (or latent traits) when there is only partial overlap of measurements across the constituent datasets. Our method, ROSETTA empirically derives a set of common latent trait metrics for each related measurement domain using a novel variation of factor analysis to ensure equivalence across the constituent datasets. The advantage of combining datasets this way is the simplicity, statistical power, and modeling flexibility of a single joint analysis of all the data. Three simulation studies show the performance of ROSETTA on datasets with only partially overlapping measurements (i.e., systematically missing information), benchmarked to a condition of perfectly overlapped data (i.e., full information). The first study examined a range of correlations, while the second study was modeled after the observed correlations in a well-characterized clinical, behavioral cohort. Both studies consistently show significant correlations >0.94, often >0.96, indicating the robustness of the method and validating the general approach. The third study varied within and between domain correlations and compared ROSETTA to multiple imputation and meta-analysis as two commonly used methods that ostensibly solve the same data integration problem. We provide one alternative to meta-analysis and multiple imputation by developing a method that statistically equates similar but distinct manifest metrics into a set of empirically derived metrics that can be used for analysis across all datasets.

## 1. Introduction

Given multiple independent cohort datasets with a common set of measurements, joint analysis of all constituent datasets is straightforward. However, when the independent cohort datasets do not have a complete set of common measurements, joint analysis is not straightforward. Methods of joint analysis can take many possible forms, such as meta-analysis or limiting the analysis design to only the few common measurements (if any), but poorly chosen joint analysis methods may trade the strengths of the constituent datasets for analytical simplicity. For example, limiting analysis to only the small subset of common measures across the datasets allows for a single mega-analysis of all the data, but that limited subset of measurements ignores much of the data at hand. Given that many data collections in biomedical studies contain correlated measurements, we believe that these correlations can be exploited to combine data using that correlated information. Our design goal was to statistically derive common latent trait metrics across datasets that will enable a data scientist to apply a single, joint analysis of all constituent datasets to: 1) retain modeling simplicity and ease of interpretation while also, 2) retain as much useful information from each dataset as is possible. By adapting the well-understood statistical procedure of factor analysis to this new task, our approach is no different than a primary analysis of a single large dataset once the data integration step has been applied (described below).

Perhaps the most common statistical approach used to integrate multiple datasets is the meta-analysis framework. Meta-analysis combines *statistical results* from each constituent dataset but does not combine *data* [1]. Although efficient in the sense that datasets need not be integrated *per se*, it can produce results that are somewhat peculiar and prone to misinterpretation. For example, the simple 95% confidence interval on a meta-analysis is not the region into which 95% of the data are expected to fall, but rather, the region into which 95% of statistical analyses are expected to fall. As a result, considerable care must be applied in characterizing and reporting results, and misunderstandings of the subtle properties of meta-analysis methods sometimes get past both authors and peer-reviewers (e.g., [2]).

We have chosen a different approach for two reasons. First, primary analysis of all the data is uniformly more powerful than meta-analysis since the total number of parameters estimated is fewer. In our procedure, each parameter has a single error term while having a greater total sample size for the estimation than any constituent dataset. In contrast, meta-analysis requires estimating parameters locally in each dataset. While the increase in power from our procedure may not always be dramatic, at least some increase in power is to be expected, typically, given the decreased number of parameters. Second, meta-analysis often subjectively combines different metrics across datasets (e.g., assuming all measurements of a physical quantity are fully equivalent regardless of the measuring device). Our procedure leverages factor analysis to specifically remove measurement error that would have otherwise been caused by assuming different measures are identical when, in fact, they are not. Indeed, our new algorithm was designed to overcome this challenge directly by deriving domain scores that are in common between the datasets.

No measurements are perfect, and all measurements are actually estimations of underlying latent traits that cannot be precisely measured or known (n.b., harkens back to Plato’s abstract objects). So long as the individual dataset measures are related to the same latent traits across all constituent datasets, it does not matter that they may not have used identical measures in all instances since datasets can be concatenated based on the common latent trait, rather than the (non-overlapping) measurements. For consistency in the newly formed mega-dataset, we hold the latent trait parameters for each domain equal across studies (outlined graphically in Figure 1).

First and foremost, ROSETTA requires that there are some measurements across the datasets in common (i.e., columns). ROSETTA does not require complete overlap of measurements nor will ROSETTA work when there is no overlap of measurements. The needed degree of overlapping measurements is discussed in more detail later. ROSETTA assumes that the datasets are independent in terms of participants (i.e., rows), though in practice, it would be possible to merge measurements into the same row prior to performing our pipeline. ROSETTA is applicable when the assumptions of latent trait modeling are generally applicable, though our algorithm does relax one key condition. For a latent trait model to be applied, the data should possess: 1) normality (or transformable to be so), 2) linear relationships between variables, 3) a large ratio of sample size to the number of measures, 4) the correlation among the measures has a latent factor structure, although the number of factors need not be hypothesized at the outset. The last two assumptions are generally not a problem for big data analysis guided by domain experts. However, implicit in factor analysis is the assumption that all subjects have all measurements. Often multiple imputation can fill in data that is missing-at-random, but large amounts of systemically missing data is not what is envisioned when multiple imputation is invoked. The goal of ROSETTA is to estimate a common set of latent traits that can be concatenated across the constituent datasets. This method has the advantage of providing complete data for downstream analysis while reducing measurement error since the analytical engine is a form of factor analysis.

## 2. Materials and Methods 

### 2.1. The ROSETTA Algorithm

**The initial factor analytic design.** The analytical foundation of ROSETTA is the mathematics behind confirmatory factor analysis, but with a modified set of inputs and, the output is not a statistical test of a hypothesis. While confirmatory factor analysis is primarily used for hypothesis testing, here we only co-opt the confirmatory factor analysis system of constraints to process the data and do not do hypothesis testing since there is no a priori model involved in ROSETTA. As such, ROSETTA is an algorithm and not a statistical test. Like all factor analytic-based methods, the goal is to represent the dataset as a lesser set of unobserved latent factors by exploiting covariance patterns [3]. Confirmatory factor analysis takes in those covariance patterns as a model whereby fit can be tested against. Here, we use the same math to impose the model as constraints for finding the linear combination of weights consistent with the imposed model. The result is a set of latent factors for downstream analysis. What makes our algorithm novel is the algorithm can be applied to datasets that have only a partial set of overlapping measures.

In practice, the correlation matrix is used in the factor analytic portion of the calculations, not the raw data; though an almost trivial observation, this step allows intervention in the algorithm at this point to perform an added data integration step. The goal then is to estimate the correlation coefficients between all measures in all datasets. If a complete matrix of all possible pairwise correlations for all measures across constituent datasets can be calculated, then the remaining steps are accomplished through standard numerical methods. 

**Calculating the overall correlation matrix.** Data from all datasets is concatenated in order to determine all Pearson correlation coefficients using available pairwise data. The result is a composite matrix of correlations for all variables, including variables that were measured in only some of the datasets. We note the following three points. 1) In practice, there is no need to build a single data structure that holds a large amount of missing data, instead, pairwise calculations can actually be done using logical intersections for calculation speed and to save memory. As a result, each entry of the correlation matrix used in the downstream factor analysis is calculated using the highest number of observations available in the total data pool and provides a de facto data integration step. 2) This simple approach ignores dataset-specific effects on correlation. One could use covariates, but we do not explore that possibility in this paper. 3) It is possible that some pairwise correlations cannot be calculated from measures available across the datasets. While it may be possible to impute the missing values in a correlation matrix, we have not studied such procedures and will not consider this contingency further here.

**Decomposition of correlations.** Eigen decomposition is performed on the correlation matrix to obtain eigenvalues and eigenvectors, which are then normalized to obtain factor loadings and communalities. Numerical routines often require positive semidefinite matrices. These can be smoothed from real data estimates if necessary using standard methods [4]. The rotation to create factor loadings is applied to the correlation matrix. This process is repeated iteratively until factors are extracted to the desired precision. Factor loadings are standardized. The correlation between factors for the regression computed score output is the key constraint to equate latent factors across studies. The number of factors needs to be specified by the user a priori or inferred from the data during the factor analysis step using standard inference methods. The ROSETTA pipeline does not perform this step for the user.

**Structural equations implement the constraints.** We then apply constraints using confirmatory factor analysis routines in an SEM framework, where the correlation between measures is constrained by the composite matrix constructed at the beginning of this procedure. Therefore, all datasets use the same correlations—this is the standard constraint in confirmatory factor analysis [3]. The correlation between factors is also held constant—a new constraint that is not customary to confirmatory factor analysis. This second constraint ensures equality of factors across datasets. Each dataset has this SEM framework applied, and the factor loadings are estimated subject to our constraints. Once those weights are derived, they can be applied to all of the constituent datasets, producing factor scores for each individual that are commensurate across datasets. Factor loadings will differ by dataset, as some measures are not identical. However, the resultant factor scores should represent the same underlying latent factor due to the constraints. These latent trait scores are then used in a primary analysis of the entire data pool.

**The ROSETTA output.** ROSETTA outputs a set of factor scores depending on how many factors were included in the analysis (and always less than the number of input measurements). These factors scores are a complete set of latent measures for each constituent dataset. The datasets can be concatenated for a single mega-analysis.

### 2.2. Simulations

In a simulation study, datasets are created based on a defined set of principles and the magnitudes of different effects (weights) for the linear combinations are chosen to reflect a given model, hereafter called a generating model. Variation in the simulated datasets comes from using random numbers to simulate the undefined sources of variation (such as measurement error). The advantage of a simulation study is that a proposed statistical method can be evaluated versus a fully specified model that has ‘true’ values to check the efficiency of the proposed statistical method (see [5] for further discussion). In contrast, results from real data analysis cannot fully establish the power of a novel method, as ground truth cannot be fully known, though real data analysis is often an important motivation for further statistical work. In terms of simulations, data were simulated by a linear combination of random numbers drawn from a normal distribution with a mean of zero and SD of one. In practice, our simulated data were generated based on a multivariate normal distribution specified by a 9x9 correlation matrix, using functions from the ‘MASS’ package [6].

The cross-trait correlation structure governed the latent structure of the nine “observed” traits and is shown as a 9x9 variance-covariance matrix (Table 1), used as input to the MASS package for simulating random numbers consistent with the model. The matrix is actually symmetric, but only the lower triangle is filled in to reduce visual clutter.

The parameters $\rho_{\mathrm{AB}},\rho_{\mathrm{AC}},\rho_{\mathrm{BC}}$ are the correlations between the three latent traits. The parameters specified by $\rho_{A1},\rho_{A2},\rho_{A3},\rho_{B1},\rho_{B2},\rho_{B2},\rho_{C1},\rho_{C2},\rho_{C3}$ are the correlations between the three different measurements of latent traits A, B and C. This type of generating model has been applied in other contexts for validating methods with underlying correlational structure [7]. 

*Control condition*. For the control condition, which acts as a benchmark for the efficiency of the analysis, we simulated a fully measured version of a large dataset (*n* = 3000 subjects) with nine quantitative measures, where each set of three measures was generated by an underlying latent trait. The three latent traits were also correlated. After simulation, the dataset was separated into three independent “studies” of *n* = 1000 for further analysis. Each study included three measures for each of the three underlying latent traits. This control condition has a complete overlap of measures across studies, and hence primary data analysis could be conducted without ROSETTA, but here, we use this full dataset condition as ground truth for benchmarking. 

*Test condition*. The test condition starts with the same data as the control condition. However, each study had a different measure removed from the dataset (observations set to missing), and thus in the ROSETTA analysis, each dataset included only two (out of the three) measures for each of the domains; further, the test condition represents an extreme case where no measure was common to all three studies for any domain. This extreme case is, therefore, informative about the robustness of the procedure. In principle, ROSETTA should return the same correlation structure across the nine measurements and the three latent traits for both the control and test conditions. 

The correlation structure was varied to form three simulations studies. The first study included a range of correlations set by the authors. Within trait correlations were fixed at 0.75. Between latent trait correlations varied in the set (0.4, 0.5 and 0.6) though we imposed this constraint $\rho_{\mathrm{AB}}=\rho_{\mathrm{AC}}=\rho_{\mathrm{BC}}$.

The second study was modeled after the observed correlations from a real dataset, the Western Reserve Reading and Math Project (WRRMP). WRRMP is an ongoing study of the cognitive development of twin children in the state of Ohio [8,9]. We chose the domains of language, reading comprehension, and single-word reading. Each of these domains have three measures. Within each domain, the dataset includes multiple quantitative measures. Correlations observed from those nine measures were used as the basis for the second simulation study.

The third study was similar to Study 1, except that between-domain and within-domain correlations were varied to facilitate developing guidelines for when ROSETTA might have more power than standard methods. Between-domain correlations were varied in the set (0.9, 0.75, 0.6, 0.45, 0.3 and 0.2) and held constant across measures in a given simulation. Within-domain correlations were varied in the set (0.1, 0.2, 0.3, 0.4) and held constant across the three factors in any simulation. In Study 3, we sought to address if the method of analysis had substantive consequences. To simulate a case status for a case-control comparison, the first measure (in domain one) was dichotomized based on values > 1 SD above the population mean (a parameter known from the simulation). Then, a t-test was performed on the seventh measure (in domain 3) comparing cases to controls. The outcome of the t-test was used to compare ROSETTA to meta-analysis and multiple imputation. 

All of the simulations presented were implemented in R, as was ROSETTA. The ROSETTA R package implementation is available at https://github.com/cwbartlett/rosetta.

## 3. Results

The advantage of simulated data is that we know what the result of statistical analysis should be since the simulated data has known, defined pairwise correlations between the variables and latent traits. We can then compare the analysis of the simulated data to the *a priori* expected result as a benchmark to determine how well the novel method is performing. We performed three simulation studies for comparison using the procedures described in the Methods section. The control condition is the complete simulated dataset, and test conditions are based on an incomplete version of that dataset. 

Simulation Study 1 showed that the ROSETTA test condition consistently had Pearson’s correlation >0.94 and usually >0.96 versus the control condition (Figure 2, Table 2 and Table 3) across all generating model correlations considered. 

This indicates that ROSETTA can capture the latent trait information across multiple datasets, allowing for missing data across studies and without complete overlap of measures.

In simulation Study 2 (Figure 3, Table 4 and Table 5), similar performance is observed. 

Given that ROSETTA can perform well with the favorable conditions in Study 1, as well as the realistic but carefully chosen conditions of Study 2, we sought to define the limits of when ROSETTA is a reasonable alternative method relative to multiple imputation and meta-analysis. Figure 4 shows the –log(*p*-value) for combinations of between and within domain correlations by method (complete data as a control condition, Rosetta, multiple imputation, and meta-analysis as comparators). When within-domain correlations are high, ROSETTA outperforms even a primary analysis of the complete data (i.e., ROSETTA outperforms the control condition with no missing data). While perhaps not immediately intuitive, this is expected because ROSETTA reduces measurement error by using multiple within-domain measurements, which increases power relative to an analysis using a single metric. As an interesting aside, imputation is sometimes more powerful than analysis of the complete data, but that might be for the same reason ROSETTA performs better, namely that imputation uses information from multiple measures, albeit in a much weaker sense than ROSETTA. Between-domain correlations have an impact on all methods, whereby strong between-domain correlations improve power. Unlike other methods, ROSETTA does have numerical issues when the within-domain correlations are too low. Note, this is a not an analytical issue, but a numerical one. Fast matrix multiplication routines in statistical software assume regularity conditions and with weak correlations, the ROSETTA correlation matrix is not well-formed by those standards. However, given that an underlying factor structure is largely absent when within-domain correlations are <0.2 (i.e., the two measures only account for <4% of the variance of each other) the assumptions of ROSETTA are not well fit.

While changes in average relative *p*-values can be a proxy for information captured by each method, *p*-values are a decision-making procedure and we sought to determine whether using ROSETTA had consequences for decision making. Table 6 shows the power of each analysis, which is the proportion of simulated replicates where the *p*-value was below alpha = 0.05. As indicated with the –log(*p*-value) graphs, ROSETTA can have higher power than a primary analysis of the complete data control condition until the within-domain correlations are <0.3. Power for all methods drops rapidly as a function of both types of correlation.

For datasets up to size *n* = 10,000, the ROSETTA procedure runs well on a standard desktop with 8 Gb of RAM in a matter of minutes and is faster than five multiple imputation instances with 50 iterations for optimization.

## 4. Discussion

An extensive body of research on data structures and algorithms for both large and small databases provides a strong foundation for research using big data [10]. The much smaller body of research on the joint analysis of datasets provides little guidance on combining large amounts of public and private datasets for big data research. When data features are not identically collected or measured, but are highly similar in nature and were intended to measure the same underlying latent trait, ROSETTA can capitalize on this information. For an illustrative example, a local study of childhood cognition would have measures of intelligence, as would equivalent public datasets on childhood cognitive development. However, there are at least twenty different tests to measure IQ found in the psychological literature that all have the claim of measuring the same underlying construct, namely intelligence. Uncritically mixing all intelligence tests would produce a noisy dataset for analysis. However, a statistical procedure that forms a latent trait derived across the different intelligence tests allows researchers to combine many datasets for a single joint analysis. The extant literature on latent traits highlights the flexibility of the approach so that it can be extended to many study designs, including time-series data and case-control data. Increases in statistical power that come from jointly deriving latent traits across many imperfectly similar datasets are likely to increase hypothesis testing power generally, and especially so when the data are highly relevant to a latent trait but possess heterogeneity across studies.

We assessed the merits of a novel application of latent trait models applicable to datasets that only partially overlap in terms of data features yet where the estimation of the underlying latent traits could be conducted accurately. We showed that empirically derived latent traits could be reliably inferred even when the constituent datasets do not have a single measure common to all of the datasets. The data integration step before rotation in the factor analysis produces a consistent estimate of the underlying latent traits.

ROSETTA is a method designed to handle a specific type of missing data—the unknown concordance and potentially missing elements of measurements across multiple data sets that were intended to measure similar features of a population, but that for various reasons did not measure them identically. A modest amount of missing-at-random data is readily handled by multiple imputation methods, and in our simulations, it performed similarly to meta-analysis even though meta-analysis was designed for this situation while imputation is generally used with far less missing information. We note that imputation could be incorporated as part of a ROSETTA analysis to fill in modest amounts of data within datasets, while still using ROSETTA across datasets. The missing data for which ROSETTA was designed can be considered a type of censored data. Censored data, which occurs when values are systematically missing, most commonly occurs because values at one end of a scale are unmeasurable. Censoring can also occur when, for example, a subset of subjects drop out of a study before completion. Censored data is an active area of research, and the solution most often occurs within the analysis model (e.g., [11]), not by changing the dataset as is the case with imputation. However, the type of missing data that ROSETTA handles differs from these contexts. ROSETTA does not replace missing data like imputation nor is ROSETTA within the downstream analysis. Instead, it seeks to replace the measurements used, with estimates of the underlying latent traits for mega-analysis. While this could be considered a kind of imputation, we would more properly classify ROSETTA as a procedure to harmonize data across datasets. 

This initial study of ROSETTA was conducted to validate the method, though there are several additional research questions that remain. For instance, it is unclear whether ROSETTA latent traits are robust to variations in sampling design from a single population. Furthermore, it is not yet known if our estimates are robust to variations in population substructure, such as studies that were ascertained in different patient populations. Research suggests the robustness of linear models under a wide variety of data inputs though there is a clear influence of outliers on accuracy. In addition, it is not known how coding time-series into the design matrix will interact with the scale-free nature of our proposed method, and in turn, influence the accuracy of the latent trait. The current study provides a foundation for pursuing these important areas of research. We also note that propensity score matching and instrumental variable estimation could be applied to improve meta-analysis or imputation-based methods beyond what is shown here. Our simulations studies lack the complexity needed for either propensity scores or instrumental variables to gain traction; further work will be necessary to define sufficiently complex models where these alternative methods can be benchmarked.

## 5. Conclusions

ROSETTA increases the utility of public data by allowing more datasets to be concatenated for a larger joint analysis. The use of a novel adaptation of latent trait analysis for the case of partially overlapping dataset features creates an empirically derived latent trait metric, common to all constituent datasets that can be analyzed similarly to a primary data analysis. This study provides information about the factors that are important to address in modeling latent traits in this context. In summary, this method produces robust results and appears to have generalizability for adoption in instances when the input data configuration is amenable.

## Figures and Tables

**Figure 1 genes-10-00727-f001:**
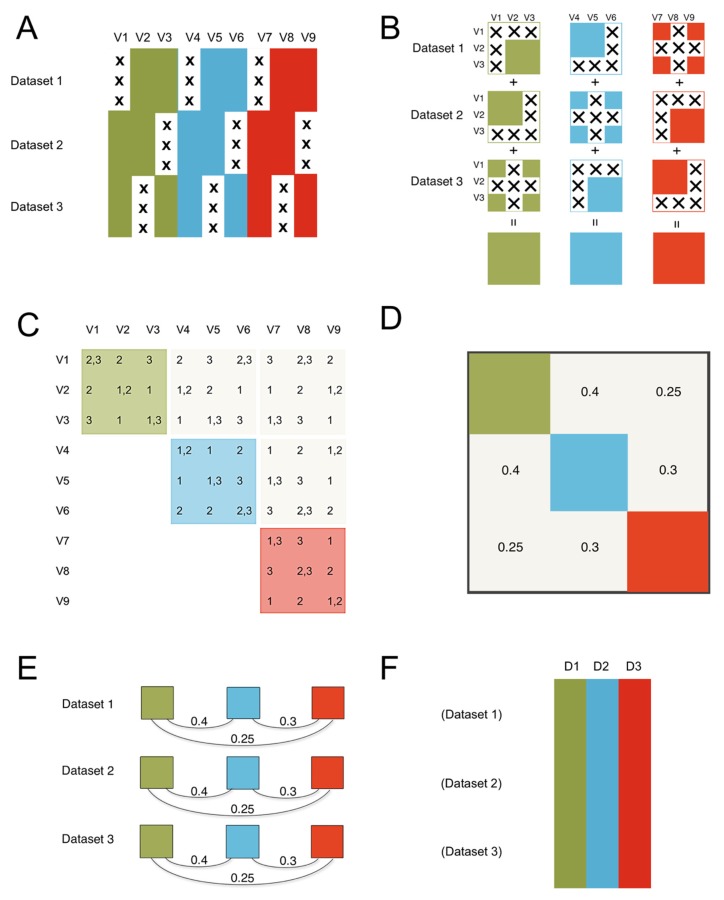
The ROSETTA flow of information. The method consists of three steps to get from the constituent datasets that are the input to ROSETTA in this example, to the final harmonized single output dataset. (**A**) The constituent datasets are concatenated and missing data is shown with an “X”. We assume the datasets are independent in terms of the rows; in biology, these are typically the different subjects that were measured. The columns are the nine measurements (V1-V9) that occurred across the three constituent datasets illustrated here. Importantly, no measurement was common to all three datasets, preventing a simple joint analysis on at least that one common measure, but the lack of complete overlap for any single measurement will not preclude using the Rosetta pipeline. We also applied color to show that the nine measurements come from three unique measurement domains. In the simulation study, we manipulate the strength of the correlations between the domains, but we expect that Rosetta is most useful when the domains have non-zero correlations. (**B**) The first step is to construct the pairwise correlation matrix for all measures across all datasets. In panel B, we show that logical intersections of the datasets allow for each domain to have a complete pairwise correlation matrix despite the pattern of missing data. (**C**) The same logic from panel B can be extended to show that the entire 9x9 matrix can be successfully estimated. The second step is to construct the geometry for the factor analysis using the 9x9 correlation matrix from step one (in panel C). The factor analysis provides a set of linear weights for combining the measurements into factor scores. (**D**) Importantly, the correlation between the factors will be used in the next step. (**E**) The third step is to apply the factor loadings and the correlations between the factors from the second step as a constraint for each constituent dataset (using the math from confirmatory factor analysis). Factor loadings are set equal to zero when a measure is not present in a given dataset, then the constraint on the correlations between the factors ensures equivalence of the factors between the datasets. While this third step is similar to the hypothesis testing of confirmatory factor analysis, where a model from one dataset is applied to a novel dataset, in ROSETTA the model was derived over all datasets and that same model is being used as a constraint when applied to each constituent dataset (i.e., Rosetta is not a hypothesis testing procedure). (**F**) The final result is a complete dataset of the domain factor scores for analysis. Rather than outputting nine variables (such as would occur with multiple imputation), Rosetta output three domain factor scores per subject (labeled D1–D3).

**Figure 2 genes-10-00727-f002:**
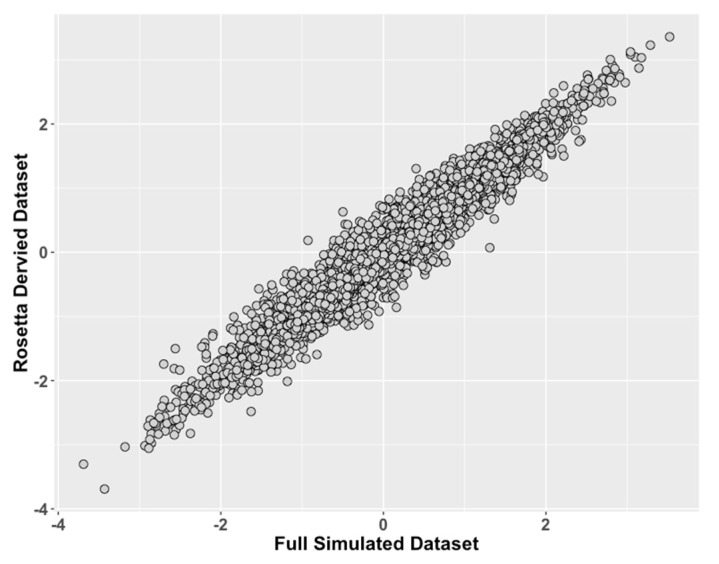
Study 1: Comparison of ROSETTA trait scores on incomplete data versus latent trait scores on complete data. We compared values of the latent traits from the full dataset condition (ground truth) on the x-axis to the incomplete matched datasets derived with ROSETTA on the y-axis.

**Figure 3 genes-10-00727-f003:**
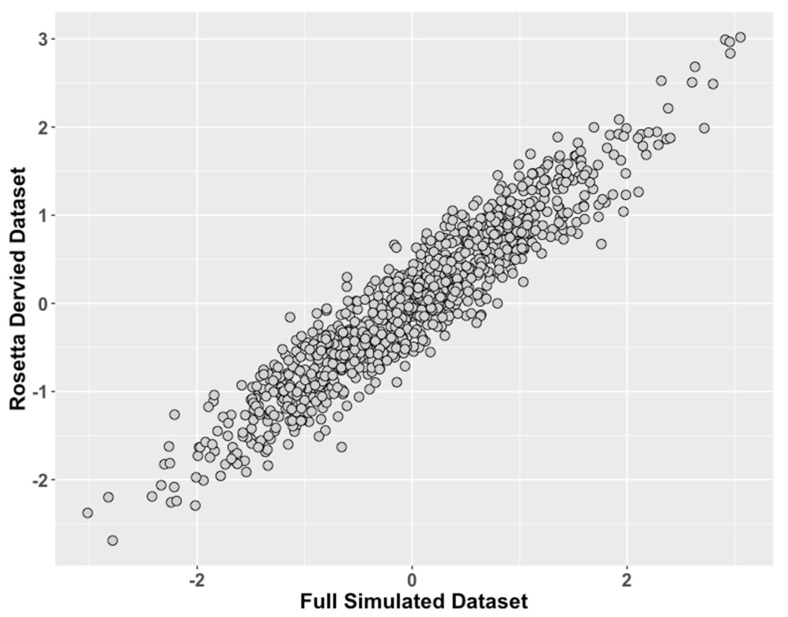
Study 2: Data Modeled After a Clinical Behavioral Dataset, Comparison of ROSETTA scores versus scores from complete data. We compared values of the latent traits from the full dataset condition (ground truth) on the x-axis to the incomplete matched datasets derived with ROSETTA on the y-axis.

**Figure 4 genes-10-00727-f004:**
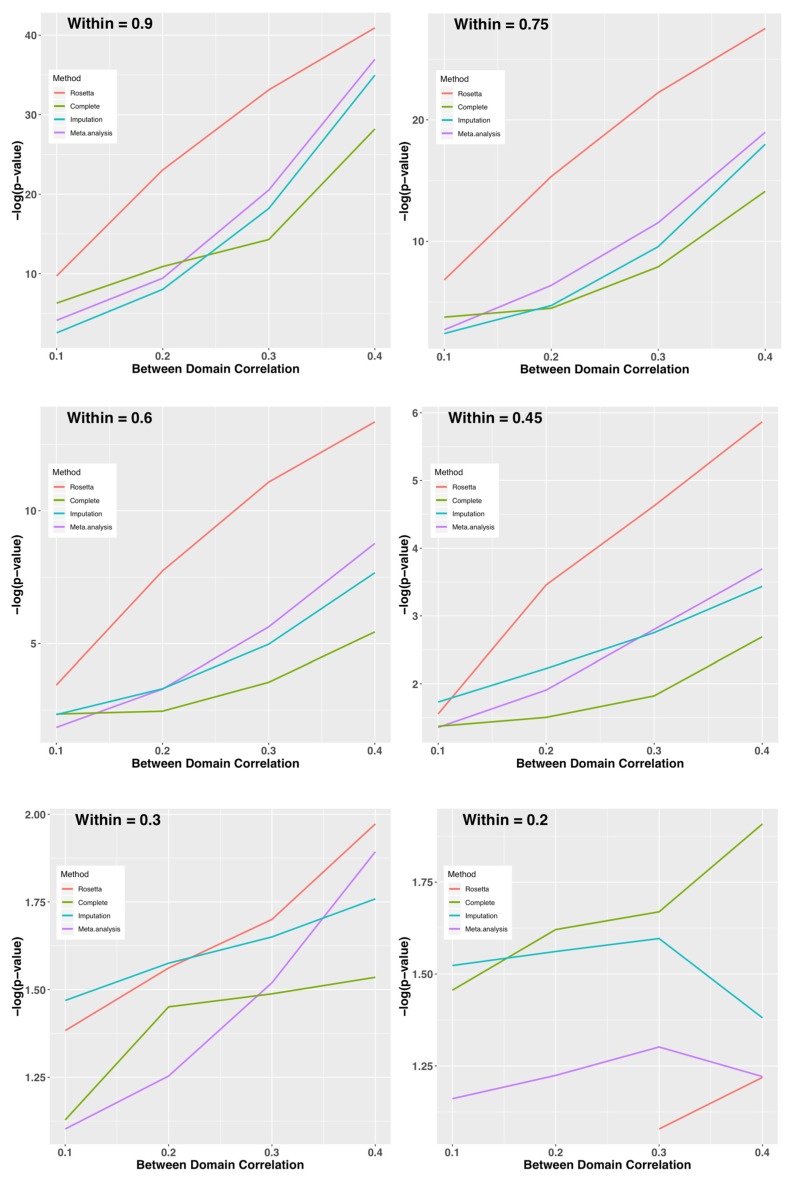
Study 3: Average –log(*p*-value) on the y-axis for Each Method by Within-Domain Correlation (panels) and Between-Domain Correlation (x-axis). Provided the within-domain correlation is >0.3 on average, ROSETTA shows a clear advantage for downstream analysis. When the within-domain correlation is 0.2 or less, the current implementation of ROSETTA runs into numerical issues and can no longer be applied.

**Table 1 genes-10-00727-t001:** Parameterized cross-correlation structure of the latent traits for simulation.

$\left[\begin{array}{ccccccccc} 1 & & & & & & & & \\ \rho_{A1} & 1 & & & & & & & \\ \rho_{A2} & \rho_{A3} & 1 & & & & & & \\ \rho_{\mathrm{AB}} & \rho_{\mathrm{AB}} & \rho_{\mathrm{AB}} & 1 & & & & & \\ \rho_{\mathrm{AB}} & \rho_{\mathrm{AB}} & \rho_{\mathrm{AB}} & \rho_{B1} & 1 & & & & \\ \rho_{\mathrm{AB}} & \rho_{\mathrm{AB}} & \rho_{\mathrm{AB}} & \rho_{B2} & \rho_{B3} & 1 & & & \\ \rho_{\mathrm{AC}} & \rho_{\mathrm{AC}} & \rho_{\mathrm{AC}} & \rho_{\mathrm{BC}} & \rho_{\mathrm{BC}} & \rho_{\mathrm{BC}} & 1 & & \\ \rho_{\mathrm{AC}} & \rho_{\mathrm{AC}} & \rho_{\mathrm{AC}} & \rho_{\mathrm{BC}} & \rho_{\mathrm{BC}} & \rho_{\mathrm{BC}} & \rho_{C1} & 1 & \\ \rho_{\mathrm{AC}} & \rho_{\mathrm{AC}} & \rho_{\mathrm{AC}} & \rho_{\mathrm{BC}} & \rho_{\mathrm{BC}} & \rho_{\mathrm{BC}} & \rho_{C2} & \rho_{C3} & 1 \end{array}\right]$

**Table 2 genes-10-00727-t002:** Study 1: Observed Correlations between ROSETTA and Full Data.

	Trait 1 – Full Data	Trait 2 – Full Data	Trait 3 – Full Data
Trait 1 - ROSETTA	0.942	0.247	0.435
Trait 2 - ROSETTA	0.262	0.993	0.409
Trait 3 - ROSETTA	0.470	0.410	0.980

**Table 3 genes-10-00727-t003:** Study 1: Expected Correlations Between Measures as Determined by Full Data.

	Trait 1 – Full Data	Trait 2 – Full Data	Trait 3 – Full Data
Trait 1 – Full data	1.000	0.252	0.451
Trait 2 – Full data		1.000	0.402
Trait 3 – Full data			1.000

**Table 4 genes-10-00727-t004:** Study 2: Observed Correlations between ROSETTA and Full Data.

	Trait 1 – Full Data	Trait 2 – Full Data	Trait 3 – Full Data
Trait 1 - ROSETTA	0.964	0.959	0.738
Trait 2 - ROSETTA	0.960	0.963	0.780
Trait 3 - ROSETTA	0.731	0.774	0.957

**Table 5 genes-10-00727-t005:** Study 2: Expected Correlations Between Measures as Determined by Full Data.

	Trait 1 – Full Data	Trait 2 – Full Data	Trait 3 – Full Data
Trait 1 – Full data	1.000	0.952	0.741
Trait 2 – Full data		1.000	0.763
Trait 3 – Full data			1.000

**Table 6 genes-10-00727-t006:** Study 3: Power of Each Method by Within-Domain and Between-Domain Correlation.

Within	Between	Rosetta-c	Truth-c	Imputation-c	Meta-c
0.9	0.4	1	1	1	1
0.9	0.3	1	1	1	1
0.9	0.2	1	1	1	1
0.9	0.1	0.92	1	0.92	0.76
0.75	0.4	1	1	1	1
0.75	0.3	1	1	1	1
0.75	0.2	1	0.88	0.88	0.92
0.75	0.1	0.96	0.8	0.48	0.64
0.6	0.4	1	0.96	0.96	0.92
0.6	0.3	1	0.88	0.84	0.84
0.6	0.2	1	0.76	0.68	0.88
0.6	0.1	0.96	0.6	0.56	0.64
0.45	0.4	1	0.8	0.72	0.76
0.45	0.3	1	0.68	0.6	0.76
0.45	0.2	1	0.68	0.52	0.6
0.45	0.1	0.96	0.4	0.28	0.52
0.3	0.4	1	0.64	0.6	0.44
0.3	0.3	1	0.48	0.36	0.44
0.3	0.2	0.96	0.36	0.36	0.48
0.3	0.1	0.8	0.36	0.48	0.48
0.2	0.4	0.32	0.4	0.24	0.32
0.2	0.3	0.17	0.28	0.48	0.48
0.2	0.2	0	0.24	0.52	0.4
0.2	0.1	0	0.24	0.48	0.44
0.1	0.4	0	0.28	0.72	0.4
0.1	0.3	0	0.24	0.2	0.36
0.1	0.2	0	0.2	0.4	0.36
0.1	0.1	0	0.2	0.6	0.36

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
