# Peer review of "Forming Big Datasets through Latent Class Concatenation of Imperfectly Matched Databases Features"

_genes, 2019, doi:10.3390/genes10090727_

Round 1

Reviewer 1 Report

This article proposes a method (Rosetta) for combining data across different studies of similar measurements (variables), replacing the values of equivalent variables observed in each study with estimates of a latent variable, which is intended to represent a common "truth" that is presumed to be observed differently in each study. The authors suggest that this method is comparable with meta-analysis, but with the advantage of allowing the estimated latent variables to be interpreted uniformly across the different studies, and therefore analyzed (presumably regardless of the chosen statistical method) as the results of a single, joint experiment.

However, this reviewer believes that meta-analysis is not the most comparable existing method. In particular, the dataset of origin can be included in an analysis as a confounding variable, allowing analysis of variance methods such as propensity score matching or instrumental variable estimation to be used to assess and control biases and covariation among the observed variables.

Furthermore, the authors do not address whether the subjects (samples) measured in each study are identical, overlapping, or distinct. For data from multiple studies, as are considered in meta-analyses, the samples measured in one study are ideally distinct from the samples measured in another study, or in some cases overlap to a small degree. It is not clear that the described method can be applied in this case. The first step in the proposed method is to compute correlations between variables across studies; however these computations can only be performed for paired measurements, ideally measurements of the same sample or to samples matched according to a credible criterion. The authors must address the degree to which must be shared across studies, or propose credible methods for pairing measurements of different samples across studies, in order to perform their analysis, and the effect of the degree of sample sharing or the matching method  on Rosetta's output (the estimated latent variable values) must be characterized.

The numerical methods are presented in conceptual dataflow illustrations (e.g. Figure 1) as well as in the Materials and Methods section (section 2.1, lines 108-145). However, neither presentation is specific enough to judge the feasibility of the method, much less to replicate results. As an example, how the correlations between "cognitive domains" (Fig. 1b) are related to the pairwise correlations (Fig. 1a) is not clearly explained. In the text (section 2.1, lines 108-117, the observed data M is not referenced after it is defined, so it is unclear how the model P = LF + e, line 111, relates to M; nor is P defined. The following paragraph, lines 122-124, completely ignores the degree of overlap in the samples (rows). Among other issues, it is questionable how relevant the pairwise correlations between samples in other studies are to a variable entirely unobserved in one of the samples; and the diagram (Fig. 1b) makes it appear as if there is one correlation coefficient between each pair of samples, when the pairwise correlations discussed would need to be combined in an unspecified manner to determine such correlation estimates. The two subsequent paragraphs, (on eigendecomposition of the correlation (matrix) and on the use of structural equations and factor analysis, are similarly vague. The "SEM framework" is mentioned (line 135), but neither explained nor annotated with a reference.

Fortunately, at the end of section 2.1, The Rosetta Algorithm, there is a link to a github repository for the code, https://github.com/cwbartlett/Rosetta. Unfortunately, at the time of this review, this repository is empty. It is therefore not possible to judge the feasibility of the method, much less to reproduce the analysis.

Section 2.2 (lines 146-194) describes two simulation studies. The first compares a "Control condition" consisting of three cohorts, each of which is fully measured on each of three variables. These three variables are assumed to be fully equivalent across all datasets, and furthermore, the scale of the correlation between paired datasets is assumed to be identical. They are also simulated as gaussian random variables, with none of the asymmetry, outliers, or other pathologies of variance encountered in real data. Only three levels of correlation, all fairly strong (minimum Pearson's rho of 0.4), were simulated. A second simulation study is described as "modeled after" a set of cognitive development data currently under collection. This data appears to be three domains, with three statistics each, collected on the same set of twin children as part of a single study. In short, neither of the simulated data sets are adequate models of the type or quantity of variation that would be observed in a meta-analysis of medical relevance.

For each simulation, results of Rosetta analysis of the fully measured data are compared to the results of Rosetta analysis of partial data. While Rosetta's ability to deal with missing data is relevant, the authors do not present any analysis of how Rosetta performs in comparison to other methods; not even any of the various imputation methods that could be used to replace missing data, nor meta-analysis of the simulated data. It is therefore not possible to assess whether Rosetta is an improvement over existing methods.

The Discussion section also fails to address the issue of how much the set of samples must overlap in order to apply Rosetta. Without addressing this issue, and providing a much broader test of how Rosetta behaves, e.g. on asymmetric as well as symmetric distributions, on more than three measures, on data previously subjected to meta-analysis, or even on data of the sort referenced in the final paragraph, "The proposed method increases the utility of public data by allowing more datasets to be concatenated for a larger joint analysis" (lines 275-276), this manuscript does not provide enough information to support Rosetta as a useful method.

This area of research is very important. A method that truly allows public data sets, acquired independently by different research groups, on different or minimally overlapping samples, and with imperfect alignment of measures reflecting a variously quantified variable such as human intelligence, results from wearable step counters or heartbeat monitors, or even measurements subject to batch effects, is badly needed for Big Data analytics. If the authors will perform a more thorough comparison of Rosetta on a much broader range of simulated data, with a more precise and reproducible description of the methods, and preferably with an example of precisely the sort of Big Data integration application Rosetta is designed to perform, I would be happy to review such a revised manuscript. Without such a major revision, however, I cannot support publication of this manuscript.

Author Response

This article proposes a method (Rosetta) for combining data across different studies of similar measurements (variables), replacing the values of equivalent variables observed in each study with estimates of a latent variable, which is intended to represent a common "truth" that is presumed to be observed differently in each study. The authors suggest that this method is comparable with meta-analysis, but with the advantage of allowing the estimated latent variables to be interpreted uniformly across the different studies, and therefore analyzed (presumably regardless of the chosen statistical method) as the results of a single, joint experiment. However, this reviewer believes that meta-analysis is not the most comparable existing method. In particular, the dataset of origin can be included in an analysis as a confounding variable, allowing analysis of variance methods such as propensity score matching or instrumental variable estimation to be used to assess and control biases and covariation among the observed variables.

RESPONSE:

            The authors agree that additional, potentially more honed, comparisons are possible.  The dominance of meta-analysis in the literature, however, dictated that we focus on that first.  For example, in searching PubMed, the primary topical search engine for the authors, there are 8 times more meta-analysis papers than propensity scores, and 71 times more meta-analysis than instrumental variables. As a first look at our current version of the method, meta-analysis appeared to be the most appropriate starting point.  We would be interested in a discussion with an expert propensity scores and instrumental variables on how to conduct a suitable comparison with our method as part of our future work.  We included text at the end of the paper to that effect.

Furthermore, the authors do not address whether the subjects (samples) measured in each study are identical, overlapping, or distinct. For data from multiple studies, as are considered in meta-analyses, the samples measured in one study are ideally distinct from the samples measured in another study, or in some cases overlap to a small degree. It is not clear that the described method can be applied in this case. The first step in the proposed method is to compute correlations between variables across studies; however these computations can only be performed for paired measurements, ideally measurements of the same sample or to samples matched according to a credible criterion. The authors must address the degree to which must be shared across studies, or propose credible methods for pairing measurements of different samples across studies, in order to perform their analysis, and the effect of the degree of sample sharing or the matching method on Rosetta's output (the estimated latent variable values) must be characterized.

RESPONSE:

            We have added in language to clarify this point, and we appreciate the reviewer pointing out this ambiguity.  Our simulations assume that the datasets do not have any overlap in subjects, and for the target application this is a reasonable assumption, even though occasionally small degree of overlap is possible.  As seen in the Figure, each dataset in our simulations had at least two measures in a given domain, which is why correlations can be calculated, as the reviewer points out.  In the Introduction, we stated that our method focuses on studies where multiple measures are taken.  This is common in biomedical experimental and cross-sectional designs, though perhaps less so for informatics studies of medical records where clinicians might only do one test or take one measurement for diagnostic purposes.  For the latter case our method would not work since correlations cannot be calculated.  We have clarified our discussion of the dataset requirements for our method. 

The numerical methods are presented in conceptual dataflow illustrations (e.g. Figure 1) as well as in the Materials and Methods section (section 2.1, lines 108-145). However, neither presentation is specific enough to judge the feasibility of the method, much less to replicate results. As an example, how the correlations between "cognitive domains" (Fig. 1b) are related to the pairwise correlations (Fig. 1a) is not clearly explained. In the text (section 2.1, lines 108-117, the observed data M is not referenced after it is defined, so it is unclear how the model P = LF + e, line 111, relates to M; nor is P defined. The following paragraph, lines 122-124, completely ignores the degree of overlap in the samples (rows). Among other issues, it is questionable how relevant the pairwise correlations between samples in other studies are to a variable entirely unobserved in one of the samples; and the diagram (Fig. 1b) makes it appear as if there is one correlation coefficient between each pair of samples, when the pairwise correlations discussed would need to be combined in an unspecified manner to determine such correlation estimates. The two subsequent paragraphs, (on eigendecomposition of the correlation (matrix) and on the use of structural equations and factor analysis, are similarly vague. The "SEM framework" is mentioned (line 135), but neither explained nor annotated with a reference.

RESPONSE:

            We reworked the methods section considerably to address these concerns.          

Fortunately, at the end of section 2.1, The Rosetta Algorithm, there is a link to a github repository for the code, https://github.com/cwbartlett/Rosetta. Unfortunately, at the time of this review, this repository is empty. It is therefore not possible to judge the feasibility of the method, much less to reproduce the analysis.

RESPONSE:

            This was an unfortunate happening and we greatly regret that the repository was not available when it was clearly needed.  We have uploaded the repository.

Section 2.2 (lines 146-194) describes two simulation studies. The first compares a "Control condition" consisting of three cohorts, each of which is fully measured on each of three variables. These three variables are assumed to be fully equivalent across all datasets, and furthermore, the scale of the correlation between paired datasets is assumed to be identical. They are also simulated as Gaussian random variables, with none of the asymmetry, outliers, or other pathologies of variance encountered in real data. Only three levels of correlation, all fairly strong (minimum Pearson's rho of 0.4), were simulated. A second simulation study is described as "modeled after" a set of cognitive development data currently under collection. This data appears to be three domains, with three statistics each, collected on the same set of twin children as part of a single study. In short, neither of the simulated data sets are adequate models of the type or quantity of variation that would be observed in a meta-analysis of medical relevance.

RESPONSE:

            The first simulation study is to establish a baseline, nothing more.  If the method can’t perform under such ideal conditions then either the method or the implementation are to be questioned. Our method passed this test.  The second simulation is actually from—what we would argue is a medically relevant—meta-analysis that we performed and are in the process of writing up for submission.  But that paper focuses on the neurogenetics of reading and language and to use that data here, while perhaps tempting as a real world example, completely derails any discussion of the method itself. Hence, we believe there is a clear need to keep the methodology separate from the first application and this concern is made all the most important by word limits.  Therefore, to avoid that very real problem of people being highly curious about the real data we analyze and less about the method, we simply modeled the simulation after a real dataset and left it at that. However, to our credit, this means we retain the complexity of real datasets, such as missing data, traits on slightly different scales, and measurement error. Perhaps the more general point being made by the reviewer is that they would be interested in a more comprehensive set of simulations to define the limits of when this method would be useful. To that end, we are fully appreciative that defining such limits would be useful to potential users of the method. We have expanded the simulations and now include a Figure 3 and a Table 5 that shows the effect of between-domain and within-domain correlations.  Note that within-domain correlations would necessarily be “fairly strong” as each constituent measure is purportedly measuring the same domain.  However, in order to provide some guidelines on when this method would be most useful, we also varied within-domain correlations.

For each simulation, results of Rosetta analysis of the fully measured data are compared to the results of Rosetta analysis of partial data. While Rosetta's ability to deal with missing data is relevant, the authors do not present any analysis of how Rosetta performs in comparison to other methods; not even any of the various imputation methods that could be used to replace missing data, nor meta-analysis of the simulated data. It is therefore not possible to assess whether Rosetta is an improvement over existing methods.

RESPONSE:

            Imputation is a data processing procedure while meta-analysis results in a p-value for making decisions.  Rosetta is more like the former; expect that instead of imputing each of the constituent trait values, each domain instead has a single factor score that represents that domain.  The trouble being that imputation, meta-analysis and Rosetta all output a different quantity so equating them is in some sense problematic.  The best we can do is take a consequentialist framework and ask, do the three procedures when coupled to a decision-making procedure in the case of imputation and Rosetta, results in different decision.  Of course the output p-values will be highly correlated, one would assume a priori, but differences in statistical power would result in differences in the false negative rate (i.e., differences in the p-values >0.05 when the null hypothesis is false).  All of the methods should have the same false positive rate (5% for p-value<0.05).  Using this consequentialist framework, we have implemented this comparison into Table 5. We believe this will help address the next concern as well. 

The Discussion section also fails to address the issue of how much the set of samples must overlap in order to apply Rosetta. Without addressing this issue, and providing a much broader test of how Rosetta behaves, e.g. on asymmetric as well as symmetric distributions, on more than three measures, on data previously subjected to meta-analysis, or even on data of the sort referenced in the final paragraph, "The proposed method increases the utility of public data by allowing more datasets to be concatenated for a larger joint analysis" (lines 275-276), this manuscript does not provide enough information to support Rosetta as a useful method.

RESPONSE:

            In response to the above, we have expanded the Introduction and Methods sections to more clearly explain the regularity conditions we view as important for the current implementation of Rosetta.  Rosetta is not as flexible as other methods, which is important to clarify, but when Rosetta conditions are met then it successfully captures more information for downstream analysis with less measurement error than either imputation or meta-analysis.  The new results section demonstrate this, which further helps to show when Rosetta is useful.  The Discussion section reiterates when Rosetta should be considered as an alternative to imputation and meta-analysis.

This area of research is very important. A method that truly allows public data sets, acquired independently by different research groups, on different or minimally overlapping samples, and with imperfect alignment of measures reflecting a variously quantified variable such as human intelligence, results from wearable step counters or heartbeat monitors, or even measurements subject to batch effects, is badly needed for Big Data analytics. If the authors will perform a more thorough comparison of Rosetta on a much broader range of simulated data, with a more precise and reproducible description of the methods, and preferably with an example of precisely the sort of Big Data integration application Rosetta is designed to perform, I would be happy to review such a revised manuscript. Without such a major revision, however, I cannot support publication of this manuscript.

RESPONSE:

            We are greatly heartened by the reviewer agreeing with the goals of the Rosetta project.  We also believe that the changes made as discussed in the responses above are adequate to make the paper truly useful.

Reviewer 2 Report

The Authors introduce a method, Rosetta, to analyze consistently several datasets with only partially overlapping features. The Authors checked their method by analyzing simulated data, and found a satisfying agreement between the data and the model.

While the manuscript is timely, and the proposed method potentially interesting and useful, the work has some important problems that must be addressed before publication.

Major:

(1) The manuscript is written in a very confusing manner. Maybe it's me, but explanations are not clear, and it is very had to understand what the goals, methods and results of the work actually are. It seems to me that the only people able to actually understand the text are the authors themselves.

Consider for instance the following sentence from the first paragraph of the introduction:
"By adapting well-understood statistical procedures to this new task, our procedure is no different than a primary analysis of a single large dataset once the data integration step has been applied (described below)"
Which statistical procedures? What kind of primary analysis? None of that is clear from context. If you are referring to factor analysis, just say it.

So, I suggest you revise the text keeping in mind that it has to be readable without too much effort by two distinct type of readers:
- Bioinformaticians used to work with biological data, but not familiar with factor analysis
- Statisticians familiar with factor analysis, but not used to work with biological data

(2) Apart from the writing style, the biggest problem I see is that while the Authors claim advantages of their method over -- over what? Please be specific in the text, see previous point -- they do not perform any concrete comparison.

For instance: "Our procedure specifically removes measurement error that would have otherwise been caused by assuming different measures are identical when, in fact, they are not. Indeed, our new algorithm was designed to overcome this challenge directly."

If this is the case, you should be easily able to show that other common methods, when applied to your simulated data, leads to wrong conclusions. Not only you have to show that your method works, you also have to assess how it performs compared to other methods.

Minor:

Line 109-112: are the X1, X2, ..., Xn variables the components of P (line 112)? Why don't you use the same letter (either X or P)

Some of the many typos I found (why don't you just use a spell checker?):
50: limited
52: partially
53: statistically
168: traits
170: Efficiency

Author Response

The Authors introduce a method, Rosetta, to analyze consistently several datasets with only partially overlapping features. The Authors checked their method by analyzing simulated data, and found a satisfying agreement between the data and the model.  While the manuscript is timely, and the proposed method potentially interesting and useful, the work has some important problems that must be addressed before publication.

Major:

(1) The manuscript is written in a very confusing manner. Maybe it's me, but explanations are not clear, and it is very had to understand what the goals, methods and results of the work actually are. It seems to me that the only people able to actually understand the text are the authors themselves.

Consider for instance the following sentence from the first paragraph of the introduction:

"By adapting well-understood statistical procedures to this new task, our procedure is no different than a primary analysis of a single large dataset once the data integration step has been applied (described below)"

Which statistical procedures? What kind of primary analysis? None of that is clear from context. If you are referring to factor analysis, just say it.  So, I suggest you revise the text keeping in mind that it has to be readable without too much effort by two distinct type of readers:

- Bioinformaticians used to work with biological data, but not familiar with factor analysis

- Statisticians familiar with factor analysis, but not used to work with biological data

RESPONSE:

            We appreciate the candid feedback and agree the onus is on us to make out points clear without making the reader work too hard.  We have annotated the revised manuscript with the location of changes.  The Methods section was unclear to Reviewer 1, so that that section has had the most revisions for clarity.  However, the Introduction and Discussion have had a number of changes for clarity, making the manuscript longer, but hopefully more enjoyable to read. 

(2) Apart from the writing style, the biggest problem I see is that while the Authors claim advantages of their method over -- over what? Please be specific in the text, see previous point -- they do not perform any concrete comparison.

For instance: "Our procedure specifically removes measurement error that would have otherwise been caused by assuming different measures are identical when, in fact, they are not. Indeed, our new algorithm was designed to overcome this challenge directly."  If this is the case, you should be easily able to show that other common methods, when applied to your simulated data, leads to wrong conclusions. Not only you have to show that your method works, you also have to assess how it performs compared to other methods.

RESPONSE:

            Taken together with Reviewer 1’s comments, the need for a more comprehensive comparison is clear and present.  We have added comparisons with multiple imputation and meta-analysis over a range of conditions.  The added power of Rosetta is evident even over a primary data analysis on complete dataset makes it clear that our claim about reducing measurement error is indeed correct.  Please see the added Table 5 and Figure 3 in the Results section.

Minor:

Line 109-112: are the X1, X2, ..., Xn variables the components of P (line 112)? Why don't you use the same letter (either X or P)

RESPONSE:

            The method section has been greatly refined in order to make sure such confusions are unlikely. 

Some of the many typos I found (why don't you just use a spell checker?):

50: limited

52: partially

53: statistically

168: traits

170: Efficiency

RESPONSE:

The original manuscript was written in a text editor for LaTex formatting (and not spell checked at that stage).  However, the conference required a Word template, so the text was put into the provided Word template where one would assume spellchecker was active.  The template had spell check turned off for reasons the first author cannot fathom.  While inexcusable to have spelling errors in a manuscript, there was at least a set of unfortunate circumstances that facilitated this travesty of copyediting.  We remedied this defect in the resubmission.

Round 2

Reviewer 1 Report

I believe you have missed the point of my comment about degree of overlap between the data sets. The most common case in which large data sets are combined is measurements of two non-overlapping cohorts on an overlapping set of measures. Cohort 1 might be measured on variables S1, S2, and A, while cohort 2 is measured on variables S1, S2, and B. A correlation coefficient can be computed between S1 and S2 in cohort 1, and between S1 and S2 in cohort 2, but it is not possible to compute correlation between cohort 1 and cohort 2 on either variable S1 or variable S2 because cohort 1 and cohort 2 do not share any samples, and therefore the values are not paired across cohorts. Thus, it is unclear how you compute "the pairwise correlation matrix for all measures across all datasets", Figure 1a. It is also not stated what V1, V2, and V3 mean in Fig. 1a; what the labels on the rows and columns should be in Fig. 1b; or what you mean by the term "clinical domain". You don't explain how many clinical domain scores are produced; is there one score per clinical domain, or one per measure within each clinical domain?

These are the kinds of omissions that must be corrected before I can assess the value of the method. It is not sufficient to ask a reader, or a reviewer, to read the code in order to understand the main properties of the method, such as how many output variables will be produced or even whether you mean Pearson, Spearman, or Kendall correlation. Likewise, you have not provided any information about the mathematical relationship between the input and output variables. Is each output variable a linear combination of the original data for the same sample? If so, what are the coefficients? Granted, they are computed in some way as latent variables, by constrained factor analysis. If k latent variables are output, are they the first k? Or does someone evaluate more, and select their favourites?

Reviewer 2 Report

The Authors' response is satisfactory, and I recommend the manuscript for publication.

Author Response

thanks